# Altered Germinal-Center Metabolism in B Cells in Autoimmunity

**DOI:** 10.3390/metabo12010040

**Published:** 2022-01-05

**Authors:** Ashton K. Shiraz, Eric J. Panther, Christopher M. Reilly

**Affiliations:** 1Department of Biomedical Sciences & Pathobiology, Virginia-Maryland Regional College of Veterinary Medicine, Virginia Polytechnic Institute and State University, 205 Duck Pond Drive, Blacksburg, VA 24061, USA; ericpanther@ufl.edu; 2Via College of Osteopathic Medicine, Blacksburg, VA 24060, USA

**Keywords:** B cells, germinal centers, BCL6, lupus, metabolism

## Abstract

B lymphocytes play an important role in the pathophysiology of many autoimmune disorders by producing autoantibodies, secreting cytokines, and presenting antigens. B cells undergo extreme physiological changes as they develop and differentiate. Aberrant function in tolerogenic checkpoints and the metabolic state of B cells might be the contributing factors to the dysfunctionality of autoimmune B cells. Understanding B-cell metabolism in autoimmunity is important as it can give rise to new treatments. Recent investigations have revealed that alterations in metabolism occur in the activation of B cells. Several reports have suggested that germinal center (GC) B cells of individuals with systemic lupus erythematosus (SLE) have altered metabolic function. GCs are unique microenvironments in which the delicate and complex process of B-cell affinity maturation occurs through somatic hypermutation (SHM) and class switching recombination (CSR) and where Bcl6 tightly regulates B-cell differentiation into memory B-cells or plasma cells. GC B cells rely heavily on glucose, fatty acids, and oxidative phosphorylation (OXPHOS) for their energy requirements. However, the complicated association between GC B cells and their metabolism is still not clearly understood. Here, we review several studies of B-cell metabolism, highlighting the significant transformations that occur in GC progression, and suggest possible approaches that may be investigated to more precisely target aberrant B-cell metabolism in SLE.

## 1. Introduction

While B lymphocytes are critical cells in autoimmunity, therapeutically targeting these cells, specifically within systemic lupus erythematosus (SLE), does not necessarily ameliorate disease [1]. Immune cells use a variety of metabolic pathways to generate energy for cell survival and to produce a plethora of effector chemicals for cellular growth, proliferation, and differentiation [2]. When immune cells are triggered by internal or extrinsic cues, metabolic reprograming occurs, moving from OXPHOS to aerobic glycolysis [3]. Due to the relationship between immune cell activity and intracellular metabolic pathways, the unbalanced immune systems in SLE patients and lupus mice models may display metabolic problems. Several studies have investigated T-cell metabolism in lupus, whereas B-cell metabolic changes have been less documented. T-cell development occurs in the thymus, where the thymic microenvironment directs differentiation and positive and negative selection. In the thymus, T cells develop specific T-cell markers, including TCR, CD3, CD4 or CD8, and CD2. T cells play a crucial role in the pathophysiology of SLE, enhancing inflammation by the release of pro-inflammatory cytokines, assisting B cells in the generation of autoantibodies, and sustaining the illness through the buildup of autoreactive memory T cells [4]. T cells in SLE show metabolic abnormalities including increased oxidative stress and mitochondrial and lipid raft abnormalities. In SLE, cellular metabolism is important in lymphocyte development and fate [5]. In B cells, OXPHOS, glucose metabolism, fatty acid (FA) metabolism, and the citric acid cycle (TCA) are altered in SLE [6,7].

Activated T cells increase glucose metabolism in order to create enough energy and to synthesize intermediate materials for cell proliferation and differentiation [8]. Both intrinsic and extrinsic metabolic parameters regulate lymphocytes [9]. At baseline, naive B cells have lower metabolic activity [7]. Compared to T cells, resting B cells have lower mitochondrial membrane potential [10]. When stimulated, B cells have an upregulation in OXPHOS, glucose uptake, fatty acid oxidation (FAO), and the TCA cycle [11,12,13]. Maciver et al. showed that B cells, like T cells, utilize glucose for activation and survival [14]. Therefore, it is plausible to say that a decrease in metabolic activity can alter B cells and the production of antibodies in SLE. Moreover, B-cell functions include antigen capture, presentation, trafficking, and antibody production, making B cells unique in function as compared to T cells. These unique properties of B cells make it critical to evaluate the metabolic targets and pathways involved in immune function. Although T cells play a critical role in the pathogenesis of SLE, B cells play a key role in activating autoreactive T cells, contributing significantly to disease pathogenesis [9].

## 2. Germinal Center Reaction

The GC is a specialized microstructure that develops in secondary lymphoid organs, where activated B cells undergo somatic hypermutation (SHM) and class switch recombination (CSR) to enhance their affinity (Figure 1). The GC produces long-lived antibody-secreting plasma cells and memory B cells [15]. GCs contains two zones: the dark zone (DZ), which contains actively dividing B cells known as centroblasts, and the light zone (LZ), which consists of non-dividing B cells known as centrocytes [15,16]. The B cells in centrocytes interact with either Follicular dendritic cells (FDCs) or antigen-specific T_FH_ cells [17]. Once the high-affinity B cells interact with FDC and T_FH_ cells, they migrate to the DZ where they undergo clonal expansion and SHM [17,18]. DZ B cells are hyperproliferative which makes them metabolically demanding [19,20]. Targeting rapamycin complex 1 (mTORC1) regulates this phenotype by promoting the anabolic program that supports DZ proliferation [21,22]. The DZ B cells receive signals from T_FH_ cells that activate mTORC1 to promote glucose uptake, cell growth, and ribosomal biogenesis [21,22]. Phosphatidylinositol 3-kinase (PI3K) signaling is also critical for the activity of centrocytes [22,23]. The GTPase R–Ras2 connects T_FH_ –derived signals to mitochondrial and glycolytic metabolism upstream of the PI3K–Akt–mTORC1 pathway [22,24]. Weisel et al. have shown that, unlike proliferating B and T lymphocytes, GC B cells conduct minimal glycolysis and are heavily reliant on FAO for ATP production [13] (Figure 2). Despite being highly active, the light zones of GCs are characterized by starvation and hypoxia [25,26]. Moreover, both hypoxia and mTORC1 activity regulate the GC B cells that are expanded in lupus and are known metabolic master regulators, which are activated in response to mitogenic signals. mTORC1 phosphorylates downstream targets promoting protein synthesis, oxidative metabolism, and glycolysis [25]. Myc is a critical transcription factor that drives the expression of genes involved in nutrient uptake, mitochondrial function, glycolysis, and glutaminolysis [27]. Within the GC, a transcription factor, Hypoxia-inducible factor 1-alpha (HIF–1α), and glycogen synthase kinase 3 (GSK3) downregulate mTORC1 signaling, which in turn enhances B-cell viability during starvation [25,26].

GSK3, a serine/threonine kinase involved in Akt signaling [28], is a vital metabolic regulator in B lymphocytes, and is critical in the creation and maintenance of GC B cells [26]. By regulating nuclear factor of activated T cells activity, active GSK3 performs a key functional role in T and B cells, promoting survival and restricting activation [29]. GSK3 function, on the other hand, is not confined to naive or quiescent cells; it is also a scaling factor for anabolic metabolism in cells with continuing or increased metabolic profiles, such as GC B cells [26]. In resting B cells, GSK3 inhibits cell growth, proliferation, and metabolic activity. GSK3 modulates B cell size, mitochondrial biogenesis, glycolysis, and ROS generation after BCR stimulation and CD40 co-activation to avoid apoptosis [26]. The PI3K pathway activates GSK3 and mTORC1 to improve glycolysis and react to fresh energy demands [30,31] (Figure 2). Glucose transporter 1 (GLUT1), also known as solute carrier family 2, facilitates glucose transport across mammalian cell plasma membranes [32]. Inhibiting PI3K could possibly reverse the TAPP-dependent increase in metabolic activity, lowering GLUT1 expression and glucose consumption, and decreasing autoantibody formation [6,33]. These findings show the importance of GSK3 and its regulators in the modulation of tolerance and aberrant signals in autoimmune B-cell growth via metabolic reprogramming. Jellusova et al. showed that the metabolic sensor, GSK3, which has been found to restrict glucose intake and preserve quiescence, is a regulator of B-cell metabolism [34]. Since GSK has a vital role in the events detailed above, further investigation is warranted to delineate the role GSK in SLE.

Activated B cells have an increase in glycolysis and OXPHOS [35] (Table 1). The two vital transcription factors, HIF-1 and c-Myc, regulate glycolysis and OXPHOS by directly binding to the promoters of genes encoding glycolytic enzymes and glucose transporters, which can influence glycolysis [20,27,36,37,38]. Once B cells enter the GC, they undergo SHM and CSR, which lead to high-affinity-antibody production. These changes are metabolically demanding; previous reports have shown that GC B cells exhibit increased glucose utilization and mitochondrial potential in comparison to those of naive B cells [20,35]. A study by Blair et al. demonstrated that the isoform of protein kinase C (PKC), which is prevalent in B cells and promotes proliferative signaling downstream of the BCR, is essential for BCR-induced glycolysis [39]. Loss of PKC-β in B cells reduces the activation of mTORC1, causing an alteration in metabolism and mitochondrial potential, which leads to aberrant GC formation and PC formation [40]. Remarkably, unlike stimulated T cells that require mTORC1 for glycolysis, stimulated B cells do not require mTORC1 activity for glycolysis [41]. However, mTORC1 signaling is vital for the positive selection of GC B cells in a T-cell-dependent manner [22]. This signaling causes the migration of B cells from the LZ to the DZ [22]. Moreover, B cells downstream of both the BCR and the costimulatory protein CD40 lack the GTPase R-Ras 2, and they fail to proliferate and migrate to the GC response [24]. R-Ras 2 deficiency in B cells inhibits the PI3K-Akt-mTORC1 pathway, slows mitochondrial DNA replication, and reduces the expression of genes involved in glucose metabolism [20]. Synergistic signals from both the BCR and CD40 in GC B cells induce c-Myc, increasing cellular metabolism and mTORC1 activation and induce the positive selection of GC B cells in the LZ to migrate to the DZ, where the hyper proliferating B cells are metabolically demanding [19,20].

In line with BCR signaling and mTORC1 activity, Choi et al. and Morel et al. have shown that AKT, a serine/threonine-specific protein kinase, plays an important role in a variety of biological functions including glucose metabolism, apoptosis, cell proliferation, and transcription [20,42]. The same group also showed that cell migration is a negative regulator of BCR signaling in GC B cells and influences mTORC1 activation [43]. T_FH_ cells are critical for the induction of CD40 signaling in GC B cells [44]. A recent study by Choi et al. used c-Myc-reporter mice to show that the CD40 signal induces c-Myc expression by LZ GC B cells in direct proportion to the amount of antigen [19,20]. Once GC B cells migrate to the DZ, c-Myc is critical for regulating DZ GC B-cell size and cell-cycle entry [45]. As GC B cells undergo cell division, their c-Myc level decreases; they recycle back to the LZ for another round of cognate Ag stimulation to start another cycle of selection [45]. Importantly, B-cell lymphoma 6 (BCL6) is a transcriptional repressor and a master regulator of GC B-cell differentiation. Only B cells expressing Bcl6 can differentiate into GC B cells [46]. Additionally, Interleukin 4 (IL-4) signaling increases Bcl6 expression through enhancing mitochondrial oxidative metabolism, which is fueled by glycolysis [46]. Haniuda, K. S. Fukao, and D. Kitamura, also demonstrated that IL-4 -mediated reprogramming of TCA cycle metabolism drives the accumulation of α-ketoglutarate (α-KG) that integrates epigenetic activation of the Bcl6 gene to induce GC B-cell differentiation [46].

GC B cells respond to oxygen sensing by balancing metabolic activity and inhibition [20]. The hypoxic environment of the LZ of GCs causes an increase in glycolysis, B-cell apoptosis, downregulation of proliferation, and disruption of immunoglobulin class switching to the pro-inflammatory IgG2 isotype by dampening the expression of activation-induced cytidine deaminase [20]. The antigen-driven selection process in the GC LZ requires precise metabolic reprogramming. Due to the hypoxic nature of the GC, the expression of HIF-1α is much greater in GC B cells than in other splenic B cells [46]. This, in turn, suppresses mTORC1 activity in B lymphoblasts in the GC’s DZ, impairing proliferation and class switching [47]. These characteristics demonstrate the importance of oxygen sensing and rapid metabolism in GC B cells. Choi et al. showed that treatment of normal or trichogenic (TC) (B6.Sle1.Sle2.Sle3) lupus-prone mice with the hexokinase inhibitor 2-deoxy-D-glucose (2DG) did not affect GC B cells or antibody production, but it significantly downregulated the activation of autoreactive GC B cells in TC mice [48]. Moreover, 2DG lowered T-and B-cell metabolism and delayed disease progression in mice. Furthermore, lupus mice treated with 2DG and metformin experienced a reversal of lupus etiology and disease severity [49,50,51] (Table 1). Thus, targeting glycolysis in the GC B cells may be a therapeutic treatment for lupus patients.

It is unclear if autoreactive GC B cells require glucose or antigen-induced T_FH_ cells for activation and differentiation. BCR-stimulated B cells do not require mTORC1 for glycolysis, which indicates that BCR-stimulated B cells are not dependent on mTORC1 [37]. It is likely that dual BCR and TLR7/9 pathways are more glycolytic and may determine the glucose requirements for autoreactive GC B cells [52,53,54]. Stimulation of the BCR has been reported to increase glucose usage by B cells in the formation of GC. Choi, et al. showed that the inhibition of glutaminolysis with 6-diazo-5-oxo-L-norleucine (DON) significantly abridged autoimmune humoral responses in both non-autoimmune and autoimmune mice. The same group showed that DON treatment significantly decreased GC formation, as well as reduced GC B cells numbers, which further shows the importance of glutamine for GC development [48]. The importance of glucose and glutamine metabolism and the contribution of the BCR, TLR signaling, and T_FH_ cell co-stimulation needs to be investigated in detail in GC B-cells in order to understand how GC B cell metabolism is being regulated.

GC B cells interact with T follicular helper (T_FH_) cells and follicular dendritic cells (FDC) [55] and undergo rapid proliferation and antigen receptor editing to enhance their affinity in a humoral immune response (Figure 1). As B cells undergo different stages of development and activation, their metabolism changes to meet the needs of each phase [9]. Haniuda et al. reported that GC B-cell development requires α-KG, and several enzymes increase cellular α-KG levels. After conversion into α-KG via glutaminase-mediated glutamate production, glutamine can enter the mitochondria and fuel the TCA during glutaminolysis (Figure 2). This pathway provides an additional substrate for the generation of the ATP molecules required for the GC B cell responses [56]. In addition to α-KG, GC B cells have an increase in oxidative metabolism which is fueled by glycolysis [46]. Furthermore, the increase in metabolism increases the transcriptional repressor BCL6, a critical regulator of the GC. The sites where B cells are selected are based on the production of antibodies with a high affinity for the antigen [57]. Several studies have shown that BCL6 inhibits cellular metabolism, including glycolysis [46,47].

## 3. B-Cell-Activating Factor and B-Cell Survival

Signaling through the BCR and B-cell-activating factor (BAFF) receptor is essential for naive B-cell survival [58]. Both of these molecules activate PI3K [59,60] and stimulate glucose metabolism and glycolysis [61] (Figure 2). Importantly, the expression of other critical glucose metabolic enzymes, including hexokinase and phosphofructokinase-1, are upregulated by BAFF signaling [60]. In lupus patients and in animal models of lupus, B cells express elevated BAFF and its homologue a proliferation-inducing ligand (APRIL), which when activated through the BAFF or APRIL receptors, increase glycolytic pathways, and produce more lupus-like autoantibodies [36,62,63,64,65]. B cells chronically stimulated with BAFF have been shown to undergo metabolic reprogramming and subsequently synthesize more antibodies, indicating that defects in the control of B-cell metabolism reconfiguration might influence the pathogenesis of SLE. A study done by Zeng et al. showed that the treatment of either Raji or Daudi human B-cell lines with rapamycin suppressed BAFF-induced proliferation and survival by activating the serine/threonine-protein phosphatase 2A (PP2A) and suppressing mTOR-mediated Erk1/2 signaling [66]. As a result, the development of antibody-secreting cells was inhibited. Likewise, B cells in lupus-prone animals overexpressed PI3K/AKT/mTOR molecules. mTORC1 was activated in the B cells of lupus-prone mice, and rapamycin inhibited B lymphocyte proliferation and survival [66,67], indicating that mTOR is associated with pathogenic autoantibody production. These findings suggest that lupus-prone B cells are strongly linked with mTORC1-dependent enhanced metabolic activity [67,68].

After an encounter with an antigen, B cells rely on glucose signaling [69]. Like activated T cells [5,70], multiple published studies have suggested that glycolysis is upregulated in activated B cells [36,71,72,73]. Several other studies have also shown that the B cells in SLE have a disrupted metabolic profile [36,74,75,76,77]. Both the dysregulated B-cell metabolism and the dysfunction of tolerogenic B-cell checkpoints have been linked to the promotion and exacerbation of disease pathology. These checkpoints have significant metabolic components and play a critical role in inhibiting the self-reactivity of B cells. One apparent mechanism underlying defective tolerogenesis and defective B-cell metabolism is increased exposure to BAFF. An increase in the level of BAFF promotes glycolysis and OXPHOS in B cells. Liu et al. showed that BAFF is critical for the survival of anergic B cells, as well as rescuing autoreactive B cells from checkpoint deletion [78,79,80]. Several clinical studies have shown that lupus patients have much higher serum BAFF than healthy individuals, which allows the autoreactive B cells in lupus patients to escape metabolic restriction [81,82,83,84].

The enzyme 5’ AMP-activated protein kinase (AMPK) plays a role in cellular energy homeostasis, mainly to increase glucose and fatty acid uptake and oxidation when cellular energy is low [85]. AMPK is activated in response to energy stress when it detects an increase in AMP. By suppressing anabolic activities that use ATP while encouraging catabolic processes that make ATP, the ATP and ADP:ATP ratios are restored and energy balance is restored [69]. It has been reported that both AMPK and mTORC1 activity were enhanced in SLE patients compared with matched healthy individuals [15,71,86]. Brookens et al. showed that AMPK maintained the balance between mitochondrial respiration capacity and homeostasis in memory B cells and plasmablast B cells. B cells deficient in AMPKα1 experienced a decline in antibody response as well as a lower number of memory B cells [70]. Moreover, memory B cells deficient in AMPKα1 displayed abnormal mitochondrial activity, reduced mitophagy, and an upregulation in lipid peroxidation [70]. AMPKα1- deficient plasmablast B cells showed signs of decline in mitochondrial respiration capacity [70]. AMPK, which acts upstream of mTOR, can be activated by metformin. Thus, AMPK may be a viable therapeutic target in SLE patients [71]. Metformin treatment also lowered blood ANA levels and inflammation in kidney and liver tissue in the mouse lupus model [71]. AMPK can also activate the NLRP3 inflammasome, which exacerbates inflammation in SLE [72]. A study by Lee et al. showed that activation of AMPK with metformin and inhibition of mTORC1 with rapamycin in B cells, inhibited GC B-cell differentiation to PCs in the Roquin mouse model [71]. Both metformin and rapamycin showed efficacy in lupus animal models [49,73,74]. 

### 3.1. Tolerogenicity in B-Cell Metabolism

Tolerogenic T cells play a critical role in inhibiting the activation of autoreactive B cells that have escaped other tolerance mechanisms; this post-activation checkpoint is metabolically driven [75]. As mentioned previously, many autoimmune disorders are characterized by the dysfunction of tolerogenic B-cell checkpoints [76,77]. Mitochondrial hyperpolarization and the generation of reactive oxygen intermediates were seen in T cells from SLE patients’ peripheral blood, together with lower levels of intracellular ATP, all of which indicated T cell mitochondrial malfunction in lupus patients [78]. It is plausible that B cells from SLE patients’ peripheral blood might act similarly to T cells and myeloid cells [79]. In lupus patients, T cells show mitochondrial synthesis and poor mitophagy [80]. Similarity B cells in SLE have larger mitochondria [81]. A study by Sumikawa et al. showed that mitochondrial dysfunction in B cells was linked to plasmablast differentiation and disease activity in SLE. They showed that enhanced mitochondrial capabilities mediated by glutamine metabolism are required for plasmablast development, which might be a therapeutic target for SLE [81].

### 3.2. Cytokine Signaling in B-Cell Metabolism

IL-4 is an anti-inflammatory cytokine that protects naive B cells from apoptosis through the up-regulation of anti-apoptotic proteins such as Bcl-x(L) and Stats-dependent regulation of glycolysis [82]. Peng, S. L., et al. used IL-4 ^−/−^ lupus-prone mice to show that IL-4 has a critical role in promoting autoantibody production in lupus nephritis. Naive B cells use extrinsic signals such as IL-4 to upregulate glucose consumption and glycolysis in a PI3K-dependent manner [82]. Although glucose metabolism is important for naive B cells, they also rely heavily on FAO to generate ATP [36]. During B-cell activation, FAO enables growth and proliferation [7]. Moreover, mitochondrial mass and PI3K–dependent glucose uptake is highly increased in stimulated murine B cells [36,37]. To sustain proliferation and antibody production, activated B cells require upregulation of the Glut1 receptor. Caro-Maldonado et al. showed that in vivo deletion of B-cell-specific Glut1 or inhibition of glycolysis led to suppression of antibody production [36]. However, Waters et al. and Doughty et al. reported that activated B cells in SLE have upregulated mitochondrial metabolism but glycolysis is not affected [35]. Instead, much of the glucose was shunted to the pentose phosphate pathway (PPP) that provided nucleotide synthesis and oxidative stress control substrates [35,37]. In the PPP, de novo FA synthesis uses glucose-derived carbons through de novo lipogenesis. Since lupus patients have overactive PPP, Perl et al. performed a metabolomics study which showed that plasma cells from lupus patients had hyperactive PPP [83,84]. FAO is used as an energy source by naive follicular B cells. Once these naive cells become activated, they switch their energy source from FAO to glycolysis [13]. After B-cell activation, glutamate provides a major substrate for mitochondrial respiration. Immune cells from lupus patients are characterized by enhanced glucose internalization and subsequent glycolysis, as well as heightened glucose oxidation by the mitochondria. Mitochondrial ATP production is reduced in patients with SLE [78], a process that has been attributed to blocked electron transport chain activity, GSH depletion, and oxidative stress [87]. Increased glucose uptake can also feed the PPP for the generation of reducing equivalents and nucleotide biosynthesis. In SLE, the two transcription factors PAX5 [88] and IKZF1 [89], suppress PPP enzymes, including G6P dehydrogenase (G6PD) and 6-phosphogluconate dehydrogenase (6PGD) [90,91]. As a result, glucose flow in B cells is skewed toward glycolysis while PPP activity is suppressed [91]. Chan et al. showed that B-cell-specific negative regulators, PAX5 and IKZF1, are important for B-cell development, as they control ATP levels in B cells, preventing leukemic transformation [90]. Moreover, the authors showed that PAX5 and IKZF1 control B-cell lineage commitment and maintained low glucose flux, presumably by regulating both Glut1 and the insulin receptor [90].

### 3.3. Oxidative Phosphorylation in B Cells

T cell mitochondrial dysfunction has been linked to SLE in previous studies [34,92,93]. However, the metabolic abnormalities of B cells in SLE are less well characterized. Since the majority of activated B cells are glycolytic [94] and rely on OXPHOS [95] any alteration in their metabolic activity can affect their function. Both toll-like receptors 4 (TLR)-4 and BCR signaling play an important role in the upregulation of glycolysis and OXPHOS and are essential for successful antibody responses in SLE [36,37,96]. Schweighoffer et al. showed that stimulated B cells with LPS led to the activation of TLR-4 signaling [97], resulting in the upregulation of OXPHOS and remodeling the mitochondria to support increased ATP production [35]. Moreover, most activated B cells employ glucose to drive the TCA and the PPP for ribonucleotide synthesis, but not glycolysis [35]. The same study discovered that eliminating glucose from B cells did not affect their activation or proliferation; however, blocking OXPHOS or glutamine metabolism completely blocked B-cell activation [35].

### 3.4. Lipid Metabolism in B Cells

Lipoprotein particles are comprised of lipids (such as phospholipids, cholesterol, and triglycerides) and apolipoproteins and are generated in the liver and gut [98]. Enriched lipids accounted for 65% of all altered metabolites in SLE patients’ serum. Moreover, recent research suggests that when mice are fed a high-fat diet, cholesterol accumulates in the spleen and lymph nodes and can increase autoantibody production in lupus-prone individuals [57]. This suggests that the metabolism of lipids is important in the immune response and pathophysiology of SLE. Lipids and cholesterol are important components of lipid rafts on the plasma membrane and are aggregated in T cells from SLE patients [99]. B cells, like T cells, in SLE, also have enhanced lipid synthesis, which has a close association with BCR activation. While both active and naive resting B cells require oxidative metabolism, naive B cells appear to rely on FAs as their primary fuel source [36]. Several studies have shown that patients with SLE and lupus-prone mouse models have higher levels of serum triglycerides [100,101]. Additionally, B cells from mice given a high-fat diet show overall immunological activation, producing more IgG and less IgM. These studies illustrate the importance of lipid synthesis in B-cell activation [102]. CD36, a class B scavenger receptor found on monocyte/macrophage cell surfaces, is involved in the detection and absorption of pro-atherogenic oxidized low-density lipoprotein (LDL) [103]. CD36 promotes cellular FA absorption, an important step in energy metabolism, and CD36 dysregulation has been shown in autoimmunity [104]. A study done by Won et al. showed that stimulation of follicular B cells with TLR and CD40 caused rapid induction of CD36 [105] (Table 1). The increase in CD36 expression allowed for a greater uptake of lipids leading to premature atherosclerosis, which is common in SLE patients [106]. These studies show the importance of lipid synthesis and CD36 dysregulation in autoimmunity. However, the role of lipid synthesis and CD36 in the B cells of SLE patients is still unknown.

GC B cells actively use FA as an energy source [13]. Blocking FA reuptake with inhibitor, etomoxir, or CD36/fatty acid transport protein (FATP) [105] can reduce the FA uptake. GC B cells showed an increase in both glycolysis and OXPHOS [35,96], and blocking glucose uptake with either 2DG or Glut1 inhibitor reduces the induction of GC autoreactive B cells in lupus-prone-mice [48]. Moreover, GC B cells have hyperactive OXPHOS [107]; blocking mitochondrial potential with metformin reduces the level of OXPHOS in GC B cells [108]. Lastly, mitochondrial ROS is much higher in pre-GC and GC than in naive B cells or plasma cells; inhibition of ROS attenuated their proliferation.

### 3.5. Antioxidant in B-Cell Metabolism

The harmful effects of ROS are minimized by the body’s various antioxidant defense systems. De novo production of thiol proteins, including glutathione and thioredoxin, are essential for antioxidant defenses, and low levels of the antioxidant molecule glutathione have been linked to a variety of clinical problems in SLE [109]. An enhanced antioxidant response may boost autoantibody formation in SLE and the maintenance of autoreactive B cells. This enhanced antioxidant protection might be a feature of B cells in SLE. Increased antioxidant capacity may change the antigen-presenting function of B cells in SLE [110]. Numerous studies show that B--T cell contact promotes autoimmunity [111,112,113]. At the immunological synapse, B cells communicate with T cells via the major-histocompatibility complex by presenting an antigen, resulting in reciprocal activation [114,115]. This immunological synapse also produces ROS, which has a major impact on T-cell activation by inhibiting phosphatases and increasing and extending intracellular signaling [116]. However, it is unclear if ROS may be influenced by the antioxidant capability of the particular APC. One in vitro study done by Kraaij et al. showed macrophages that produce ROS tend to generate new regulatory T cells (Tregs) in a ROS-dependent manner [117]. Other research suggests that ROS stabilize Foxp3 expression by activating SUMO-specific protease 3 (Senp3) and retaining Bach2 in the nucleus, reducing T-effector gene expression and preserving Treg stability [118]. Moreover, Akkaya et al. showed that metabolic activation of murine B cells with α-IgM lead to the accumulation of ROS and mitochondrial dysfunction [75]. The presence of B cells with increased antioxidant capacity may minimize the amount of ROS that a docked Treg encounters, reducing Foxp3 stabilization and, hence, the loss of Treg stability or the induction of new Tregs. Therefore, ROS-mediated oxidative damage results in the generation of harmful byproducts such as aldehydic compounds and the formation of protein adducts. As a result of this effect, they are highly immunogenic, resulting in pathogenic antibodies in SLE patients [119].

### 3.6. B-Cell Differentiation and Plasma Cells

As mentioned previously, the GC reaction generates both affinity-matured memory B cells and antibody-secreting PCs. PCs, as terminally differentiated effector B cells, are likely to have specific metabolic requirements. Antibodies secreted by autoreactive short-lived plasmablasts and long-lived plasma cells (LLPCs) can contribute significantly to autoimmune pathogenesis [110,120]. PCs require a large amount of glucose for immunoglobulin glycosylation, and a small portion of this glucose is catabolized into pyruvate that must be imported into the mitochondria for the survival of LLPCs. LLPCs consume more glucose and mainly use glucose to glycosylate antibodies. Both short-lived and LLPCs upregulate their Glut1 receptor to increase the uptake of glucose and utilize it for antibody glycosylation [121,122]. Although glycolysis is important for the production of ATP, most of the glucose utilized by PCs is used for antibody glycosylation via the hexosamine pathway [96,122]. During metabolic deficiency, PCs can redirect glucose from glycosylation to glycolysis, with the resulting pyruvate being delivered into the mitochondria to re-establish equilibrium [122]. Interestingly, increased glycolysis via hexokinase-2 overexpression favors PC differentiation rather than self-renewal, whereas suppression of glycolysis results in reduced PC differentiation or survival [121,123]. There is clear evidence of a relationship between immunoglobulin synthesis and bioenergetic metabolism. Moreover, there is not much known about memory B-cell metabolism, although these cells are believed to be metabolically quiescent.

The two important PC differentiation transcription factors, Blimp1 and XBP1, cause significant endoplasmic reticulum (ER) enlargement during PC differentiation [124]. High amounts of immunoglobulin synthesis results in the buildup of misfolded proteins and ER stress. PCs use both antioxidant mechanisms and the unfolded protein response to maintain metabolic balance, which inhibits mRNA translation while improving protein folding capacity and the ER’s ability to remove misfolded proteins [125,126]. Blimp1 also promotes oxidative metabolism, which causes metabolic remodeling during PC differentiation [96]. The PC’s respiratory capacity depends on OXPHOS, which is fueled by fatty acids, with pyruvate acting as its substrate. Lam et al. discovered that both mouse and human LLPCs utilize long-chain fatty acids as their principal fuel source, resulting in increased mitochondrial respiratory capacity [122]. Since all of the aforementioned events are characteristics of SLE immune cells, the autoantibody-producing PCs, XBP1, Blimp1, and OXPHOS may be potential therapeutic targets for SLE.

### 3.7. Amino Acids and B-Cell Metabolism

Amino acid metabolism is important in both T-cell and B-cell metabolism, in addition to glycolysis and FAO synthesis [6,127,128]. Many activities involved in cell proliferation, development, and function rely on amino acid metabolism. The α-amino acid, glutamine, plays a vital role in the protein biosynthesis of immune cells. T and B lymphocyte immunological responses require glutamine metabolism, as both activated T and B lymphocytes utilize significant amounts of glutamine in response to antigen receptor stimulation [129]. PCs utilize amino acid metabolism to synthesize antibodies. Many amino acids are formed from glutamine through glutaminase and transaminase enzymatic reactions; glutamine can also act as a substrate for oxidative metabolism [121]. PCs use amino acids to synthesize antibodies and this requires the activation of the energy-regulating kinase complex, mTORC1, which is vital for PC differentiation and antibody production [121,130]. A study done by Brookens et al. showed that AMPK promotes metabolic homeostasis of PCs but inhibits antibody production of those cells [70]. Tryptophan is one of the nine essential amino acids, meaning it is not produced by humans and must be obtained through food supplementation and breakdown by gut bacteria. The metabolism of tryptophan is reliant on intestinal microbiota absorption and is intimately tied to autoimmunity [131]. Previous research found decreased amounts of tryptophan but greater levels of kynurenine in the sera of lupus patients [132].

## 4. Conclusions

The exact B-cell metabolism and signaling network in autoimmunity has not been fully elucidated, as there is a paucity of knowledge in the metabolism of Bregs, PCs, and memory B cells in SLE. Understanding the metabolic pathways that govern B cells regarding their development, cytokine synthesis, and antibody release is crucial for understanding B-cell pathophysiology. Because glucose metabolism and the PPP are required for B-cell clonal proliferation, plasma-cell differentiation into antibody-secreting cells, and cytokine generation [37], it is feasible that therapeutics may be administered to specifically target these steps. FA biosynthesis is increased in antibody-secreting cells and acts to maintain the endoplasmic reticulum expansion that is required for antibody production [133]. There is a question as to what effects an increased free FA uptake has on B-cell phenotypes. There is evidence that suggests B-cell metabolism, which is tightly regulated in healthy patients, is lost in autoimmunity.

Recent studies in lymphocyte metabolism have demonstrated that there are altered pathways in both T-and B-cell subsets in autoimmunity. T lymphocytes develop into pro-inflammatory or regulatory cells based on the intricate selection of dominant metabolic pathways. There is also a unique prevalence of particular metabolic pathways in resting naive cells, memory cells, and plasma cells in B lymphocytes. Using genetically altered mice with cell-specific defects in selected metabolic enzymes, researchers have uncovered pathways critical for differentiation and activation of lymphocytes. These investigations have demonstrated the relevance of mediators such as acetyl-CoA, glycolysis, and OXPHOS enzymes, in the differentiation of functional subsets.

## Figures and Tables

**Figure 1 metabolites-12-00040-f001:**
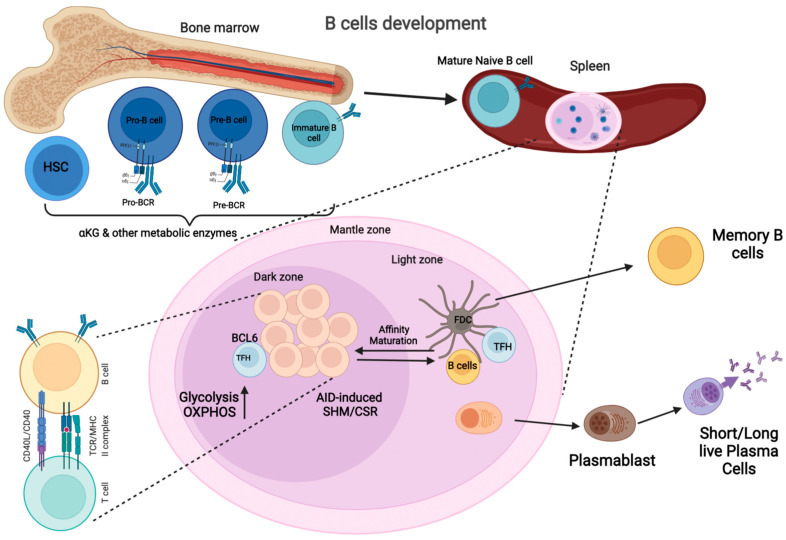
B-Cell Development Pathways: B-cells development originates from the hematopoietic stem cells in the bone marrow (BM). The early stages of B-cell development are antigen-independent and require the formation of multiple intermediary precursor cells, including Pro-B cells, Pre-B cells, which grow into immature B cells, from B lymphocyte progenitor cells. B cells undergo immunoglobulin gene rearrangement throughout these phases of development, culminating in the creation of a mature B cell receptor (BCR) capable of binding to an antigen. This is followed by a selection procedure using BCR editing or clonal deletion, which are intended to exclude autoreactive immature B cells. The majority of immature B cells that survive negative selection leave the BM and move to a secondary lymphoid organ, such as the spleen, where they enter GC reactions after antigen-dependent activation. In the DZ of the GC, they undergo SHM and CSR to increase their affinity for specific antibodies; they then develop into memory B cells or long-lived, antibody-secreting plasma cells (Created with BioRender.com, accessed on 21 December 2021).

**Figure 2 metabolites-12-00040-f002:**
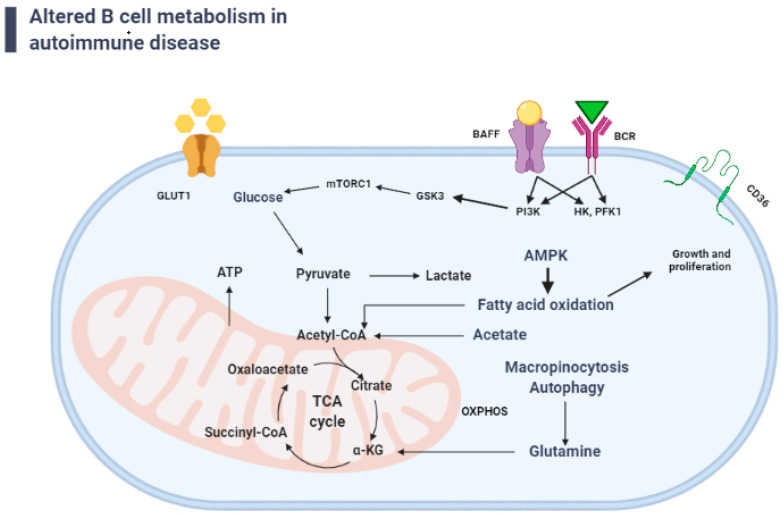
Autoreactive B cell’s energy metabolism: The glycolysis route, the tricarboxylic acid (TCA) cycle, and the oxidative phosphorylation (OXPHOS) system all produce ATP to fulfill cellular energy demands. Glucose, the primary energy source, enters the cell via glucose transporters (GLUT-1) and is transformed to pyruvate (PYR). PYR can take one of two paths: it can be transformed into lactate and exit the cell, or it can enter the mitochondrion and create Acetyl coenzyme A. (AcCoA). Signaling through the BCR and B-cell-activating factor (BAFF) is essential for naive B cell survival, as both of these signals activate PI3K. Importantly, the expression of other critical glucose metabolic enzymes including HK and FPK-1 (hexokinase and phosphofructokinase-1) are upregulated by BAFF signaling. Fatty acid oxidation (FAO) is also critical for the B-cell growth and proliferation (Created with BioRender.com, accessed on 21 December 2021).

**Table 1 metabolites-12-00040-t001:** Altered immunometabolism, significance, and potential treatment for glycolysis, FAO, and OXPHOS. This Table was adapted from the article by Wilson, C.S. and Moore, D.J. [79].

Altered Immunometabolism	Significance	Potential Treatment	
Enhanced lipid uptake due to the upregulation of CD36/fatty acid transport protein	Activation of inflammasomes	Etomoxir CD36 blocker	A
Enhanced glucose uptake	Enhanced glycolysis in naive B cells and altered glycosylation and function on immunoglobulin	2DG or Glut1 inhibitors	B
Increase mitochondrial mass and function	Increased inhibition of regulatory phosphatase due to mitochondrial ROS	Metformin	C
ROS resistance	Resistance to metabolic stress	ROS inhibitor	D

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
