# Peer review of "Altered Germinal-Center Metabolism in B Cells in Autoimmunity"

_metabolites, 2022, doi:10.3390/metabo12010040_

Round 1

Reviewer 1 Report

The review article entitled “Altered germinal center metabolism in B cells in autoimmunity” by Shiraz et al. summarized recent findings regarding B cell metabolism related with normal and autoimmune diseases.  The manuscript is well written and easy to read.  I have few comments.

  1. In line 39, it states “the metabolic activity of naïve B cells is very low…:, whereas in line 94, it states “ General, naïve B cells have an active metabolism”. These two statements seem contractional.   
  2. In the B cell development section (lines 63-92), authors state autoreactive B cells are eliminated by negative selection though deletion or receptor editing. Actually, some autoreactive B cells escape negative selection by becoming  anergy B cells.  (Nature Reviews Immunology volume 7, pages633–643 (2007)).

Author Response

Dear reviewer, 

We would like to thank the reviewers for their comments and suggestions regarding our manuscript “Altered germinal center metabolism in B cells in autoimmunity. We believe we have addressed each comment and made corrections in our manuscript where warranted.  Below is a point-by-point response to each comment.  Also, attached please find one copy of the manuscript in review mode that shows the changes and clean copy of the manuscript.  

  1. In line 39, it states “the metabolic activity of naïve B cells is very low…:, whereas in line 94, it states “ General, naïve B cells have an active metabolism”. These two statements seem contractional.   

Response: We revised the sentence accordingly:

Please see lines 42-43

  1. In the B cell development section (lines 63-92), authors state autoreactive B cells are eliminated by negative selection though deletion or receptor editing. Actually, some autoreactive B cells escape negative selection by becoming anergy B cells.  (Nature Reviews Immunology volume 7, pages633–643 (2007)).

Response: We revised the sentence accordingly: Please see lines 218-219

​Ashton K Shiraz

D.V.M./Ph.D. Candidate

Department of Biomedical Sciences and Pathobiology

Virginia-Maryland Regional College of Veterinary Medicine

Virginia Polytechnic Institute and State University, Blacksburg Virginia

Reviewer 2 Report

  It is an interesting article reviewing investigations of B cell metabolism, highlighting the significant transformation occurring in progression of germinal center. The authors suggested more approaches for understanding dysregulated B cell metabolism to identify therapeutic targets for B cell-mediated autoimmunity in systemic lupus erythematosus including human patients and veterinary victims.

  The manuscript is well written in English, and the contents might be relevant to the clinical application. There are some minor suggestions as follows.

  1. In the Abstract lines 9 and 10, please modify the sentence as follows. “B lymphocytes are --- secretion of cytokines and chemokines, and autoantigens presentation”. In addition, please remove references in the Abstract section.
  2. In the Table 1, D was missing in legends. Please remove references in the Table legends.
  3. In the Conclusion section, please abbreviate the length as possible.
  4. In the Reference section, please check the correctness of no. 2 and no. 3. No. 2 should be Front Immunol, 2021. 12(735463), and no. 3 should be Front Immunol, 202. 12(681105).

Author Response

Dear reviewer, 

We would like to thank the reviewers for their comments and suggestions regarding our manuscript “Altered germinal center metabolism in B cells in autoimmunity. We believe we have addressed each comment and made corrections in our manuscript where warranted.  Below is a point-by-point response to each comment.  Also, attached please find one copy of the manuscript in review mode that shows the changes and clean copy of the manuscript.  

  1. In the Abstract lines 9 and 10, please modify the sentence as follows. “B lymphocytes are --- secretion of cytokines and chemokines, and autoantigens presentation”. In addition, please remove references in the Abstract section.

Response: Thank you very much for your previous comments that helped us improve this manuscript. Lines 9 and 10 were changed and the references in the abstract have been removed.

  1. In the Table 1, D was missing in legends. Please remove references in the Table legends.

Response:  Revised accordingly. D was removed in the table legend.  Please see lines 297-298

  1. In the conclusion section, please abbreviate the length as possible.

Response: Thank you, the conclusion is briefly summarized.

  1. In the Reference section, please check the correctness of no. 2 and no. 3. No. 2 should be Front Immunol, 2021. 12(735463), and no. 3 should be Front Immunol, 202. 12(681105).

Response: Thank you so much for the reminder. We revised accordingly. The references have been changed to their appropriate numbers. Please see lines 403 and 404

​Ashton K Shiraz

D.V.M./Ph.D. Candidate

Department of Biomedical Sciences and Pathobiology

Virginia-Maryland Regional College of Veterinary Medicine

Virginia Polytechnic Institute and State University, Blacksburg Virginia

Reviewer 3 Report

I read with interest the manuscript entitled “Altered germinal center metabolism in B cells in autoimmunity” by Ashton Shiraz, et al that is intended to be published in IJMS journal.

Major concerns:

Need references: Although B lymphocytes are critical cells in autoimmunity, therapeutically targeting 29 these cells, specifically within systemic lupus erythematosus (SLE), does not necessarily 30 ameliorate disease

Here some description of T-cell function before focusing on its metabolism: In SLE, cellular metabolism is important in lymphocyte development 31 and fate [1].

Not understandable: However, few studies have tried to comprehensively address either normal 32 B cell metabolism or autoreactive B cells in autoimmunity

Further is repetitive: Further investigations are needed to further define the role 35 of metabolism in autoreactive B cells in SLE

the metabolic activity of naïve B cells is very low which is an important factor 39 in preventing aberrant activation. The low metabolic activity is a physiological characteristic of these cells, not specially an obstacle for aberrant activation?

Not understandable: and its dysregulation in many has been shown in autoimmunity

CD36 rapidly induces 48 in follicular B cells in vitro following, should be better: CD·& is rapidly induced in …

Since B lymphocytes are multifunctional (antibody 52 production, activation of T cells through antigen presentation, and cytokine production): ??? activation of T cells through antigen presentation?????

I do not continue reading the manuscript, it must be rewritten helped by expert medical writers and experts in the field.

Author Response

Dear reviewer, 

We would like to thank the reviewers for their comments and suggestions regarding our manuscript “Altered germinal center metabolism in B cells in autoimmunity. We believe we have addressed each comment and made corrections in our manuscript where warranted.  Below is a point-by-point response to each comment.  Also, attached please find one copy of the manuscript in review mode that shows the changes and clean copy of the manuscript.

Need references: Although B lymphocytes are critical cells in autoimmunity, therapeutically targeting 29 these cells, specifically within systemic lupus erythematosus (SLE), does not necessarily 30 ameliorate disease

Response: Thanks for your kind reminders. We revised the sentence as follows:  The sentence contains the appropriate reference.

Here some description of T-cell function before focusing on its metabolism: In SLE, cellular metabolism is important in lymphocyte development 31 and fate [1].

Response: Thanks for your kind reminders.  Please see lines 37-38

Not understandable: However, few studies have tried to comprehensively address either normal 32 B cell metabolism or autoreactive B cells in autoimmunity

 Response: Thanks for your kind reminders.  The sentence has been corrected

Please see lines 44-46

Further is repetitive: Further investigations are needed to further define the role 35 of metabolism in autoreactive B cells in SLE

 Response: Thank you; sentence have been fixed accordingly

the metabolic activity of naïve B cells is very low which is an important factor 39 in preventing aberrant activation. The low metabolic activity is a physiological characteristic of these cells, not specially an obstacle for aberrant activation?

Thank you; sentence have been fixed accordingly.  Please see line 42

Not understandable: and its dysregulation in many has been shown in autoimmunity

 Response: Thanks for your kind reminders. The sentence has been corrected. Please see line 283-284

CD36 rapidly induces 48 in follicular B cells in vitro following, should be better: CD·& is rapidly induced in …

  The sentence has been reconstructed please see lines 284-286

Since B lymphocytes are multifunctional (antibody 52 production, activation of T cells through antigen presentation, and cytokine production): ??? activation of T cells through antigen presentation?????

Response: Thank you very much for your comments however, we would like to ask for some specific clarification

I do not continue reading the manuscript, it must be rewritten helped by expert medical writers and experts in the field.

​Ashton K Shiraz

D.V.M./Ph.D. Candidate

Department of Biomedical Sciences and Pathobiology

Virginia-Maryland Regional College of Veterinary Medicine

Virginia Polytechnic Institute and State University, Blacksburg Virginia

Round 2

Reviewer 3 Report

This is a second round of the manuscript entitled “Altered germinal center metabolism in B cells in autoimmunity” by Ashton Shiraz, et al that is intended to be published in IJMS journal. In the first round I really could not read the article because the narrative construction. Now, it has greatly improved. I enjoyed the reading of the MS and I think it is ok for publishing. Authors explains the B and T cell functions through their metabolic activity. It gives a lot of interesting information in this issue.

However, there is a confusion in the references in the text. And in some paragraphs, there are some inconsistences, as I attach below. They should be corrected.

Line 92, there is some problem with references: GSK3, serine/threonine kinase involved in Akt signaling, is vital metabolic regulator 92 in B lymphocytes, and is critical in the creation and maintenance of GC B cells [85, 99]. 93 By regulating nuclear factor of activated T cells (NFAT) activity, active GSK-3 performs a 94 key functional role in T and B cells, promoting survival and restricting activation [100]. 95 GSK-3 function, on the other hand, is not confined to naive or quiescent cells; it is also a 96 scaling factor for anabolic metabolism in cells with continuing or increased metabolic pro-97 files, such as GC B cells [85].

Line 240 is erroneous: The majority of activated B cells, like CD4+ T cells, are glycolytic, but the precise process is still unknown [58].

Line 243, not understandable: Two key factors that exacerbate and promote SLE are dysregulated B cell metabolism and dysfunction of tolerogenic B cell checkpoints

Line 245, not understandable: It is increased to increase glucose and fatty acid uptake and oxidation when cellular energy is low.

Author Response

Dear Reviewer,

Please find the attached response.

Thank you so much,
Ashton K Shiraz
D.V.M./Ph.D. Candidate
Department of Biomedical Sciences and Pathobiology
Virginia-Maryland Regional College of Veterinary Medicine
Virginia Polytechnic Institute and State University, Blacksburg Virginia
